 **eLIFE**

# Natural antisense transcripts regulate the neuronal stress response and excitability

**Xingguo Zheng[1], Vera Valakh[2], Aaron DiAntonio[2,3], Yehuda Ben-Shahar[1]\***

[1]Department of Biology, Washington University in St. Louis, St. Louis, United States; [2]Department of Developmental Biology, Washington University School of Medicine, St. Louis, United States; [3]Hope Center for Neurological Disorders, Washington University School of Medicine, St. Louis, United States

**Abstract** Neurons regulate ionic fluxes across their plasma membrane to maintain their excitable properties under varying environmental conditions. However, the mechanisms that regulate ion channels abundance remain poorly understood. Here we show that *pickpocket 29* (*ppk29*), a gene that encodes a *Drosophila* degenerin/epithelial sodium channel (DEG/ENaC), regulates neuronal excitability via a protein-independent mechanism. We demonstrate that the mRNA 3′UTR of *ppk29* affects neuronal firing rates and associated heat-induced seizures by acting as a natural antisense transcript (NAT) that regulates the neuronal mRNA levels of *seizure* (*sei*), the *Drosophila* homolog of the human *Ether-à-go-go* Related Gene (hERG) potassium channel. We find that the regulatory impact of *ppk29* mRNA on *sei* is independent of the sodium channel it encodes. Thus, our studies reveal a novel mRNA dependent mechanism for the regulation of neuronal excitability that is independent of protein-coding capacity.

## Introduction

The neuronal action potential is sensitive to abrupt changes in environmental temperatures (*Peng et al., 2007*; *Buzatu, 2009*). Thus, the failure of neurons to adjust their physiological properties in response to a fast rise in temperature can lead to neurological disorders such as febrile seizures (*Bassan et al., 2013*). Previous theoretical and experimental studies suggested that one of the main mechanisms for maintaining normal neuronal excitability, circuit integrity, and behavioral robustness under varying environmental temperatures depends on changes in the abundance and membrane half-life of various voltage-dependent ion channels (*Marder and Prinz, 2003*; *O'Leary et al., 2013*; *Rinberg et al., 2013*; *Rosati and McKinnon, 2004*; *Tang et al., 2010*, *2012*). However, the actual molecular mechanisms that mediate these processes are largely unknown.

Several whole genome transcriptomics studies revealed that natural antisense non-coding transcripts (NATs) are prevalent in eukaryotes (*Lapidot and Pilpel, 2006*; *Okamura et al., 2008*). Although the function of the majority of NATs is still unknown, evidence suggests that at least some *cis*-NATs are likely to act as regulatory RNAs of protein-coding transcripts (*Borsani et al., 2005*; *Okamura et al., 2008*; *Watanabe et al., 2008*), including a recent report about a non-coding NAT that regulates a neuronal potassium channel (*Zhao et al., 2013*). Furthermore, some NATs have been shown to play a role in the physiological response to various stresses in plants (*Borsani et al., 2005*; *Katiyar-Agarwal et al., 2006*). Whether NATs play a role in the post-transcriptional regulation of ion channels and neuronal excitability was unknown.

## Results and discussion

### *ppk29* and *sei* are convergently transcribed ion channels

The response of neurons to acute heat stress is likely to require rapid changes in ion channel functions. We hypothesized that NATs play a role in the posttranscriptional regulation of ion channel

**\*For correspondence:** benshahary@wustl.edu

**Competing interests:** The authors declare that no competing interests exist.

**eLife digest** Neurons communicate with one another via electrical signals known as action potentials. These signals are generated when a stimulus causes sodium and potassium ion channels in the cell membrane to open, leading to an influx of sodium ions, followed by an efflux of potassium ions. Changes in temperature affect the rate at which ion channels open and close, and thus affect how easy it is for a stimulus to trigger an action potential. In response to a sudden rise in temperature, neurons must adjust the number of ion channels in their membranes to ensure that they do not become hyperexcitable, which could result in epilepsy.

Now, Zheng et al. have revealed one possible mechanism for how neurons do this. In the fruit fly, *Drosophila*, a gene for a potassium channel is found on the same chromosomal location as a gene for a sodium channel, and some of the genetic elements that regulate the expression of these two genes even overlap. However, the genes are on opposite strands of the DNA double helix. This means that when the genes are transcribed to produce molecules of messenger RNA (mRNA), which is usually single stranded, some of the mRNA molecules will pair up to form double-stranded mRNA molecules. This is significant because such RNA 'duplexes' have been shown to inhibit the translation of conventional single-stranded mRNA molecules into proteins, or to lead to their complete degradation.

Zheng et al. found that flies with mutations in the potassium channel gene display seizures in response to sudden changes in temperature. However, insects with mutations in the sodium channel gene are not affected because, surprisingly, they have a higher than expected number of potassium channels. It turns out that the mutant sodium channel mRNA molecules are unable to form RNA duplexes with potassium channel mRNA molecules: these duplexes would normally limit the number of potassium channels so, in their absence, the number of potassium channels increases, and this protects the flies from seizures.

Zheng et al. also uncovered a novel mechanism by which mRNA molecules can regulate gene expression independent of their role as templates for proteins. Further work is required to determine whether this mechanism is also present in other organisms, including humans.

function in response to stress. Therefore, we screened the well-annotated genome of the fruit fly *Drosophila melanogaster* to identify known excitability-related ion channels that might be regulated by endogenous NATs.

Using this approach, we found that the gene *seizure* (*sei*), which encodes the sole fly homolog of the human *Ether-à-go-go* Related Gene (hERG) inward rectifying K$^+$ channel (**Titus et al., 1997**; **Wang et al., 1997**), is located downstream of the degenerin/epithelial sodium channel (DEG/ENaC) *ppk29* (**Liu et al., 2012**; **Thistle et al., 2012**; **Zelle et al., 2013**). The two genes are convergently transcribed on opposite DNA strands, and have complementary 3'UTRs that overlap by 88 nucleotides, which we confirmed by fully sequenced cDNAs deposited in NCBI and 3'RACE analysis (**Figure 1A**). The two opposing physiological functions of *sei* and *ppk29* (K$^+$ and Na$^+$ channels respectively), and the realization that their transcripts could form natural sense/antisense RNA duplexes (**Katayama et al., 2005**; **Czech et al., 2008**) led us to hypothesize that the mRNAs of *sei* and *ppk29* may regulate each other via the formation of natural endogenous dsRNAs. Since mRNA-dependent interaction between *sei* and *ppk29* requires that the two genes will be co-transcribed we first analyzed expression data from the modENcode (**Cherbas et al., 2011**) and the FlyExpress (**Robinson et al., 2013**) projects. Although previous studies suggested that *ppk29* function might be a sensory-specific (**Liu et al., 2012**; **Thistle et al., 2012**), our analysis revealed that *sei* and *ppk29* are co-expressed in neuronal cell lines (**Figure 1—figure supplement 1A**) and are both enriched in the fly central nervous system (**Figure 1—figure supplement 1B**). *In situ* hybridization in the fly brain also demonstrated neuronal co-expression (**Figure 1B–D**). Furthermore, we used cell-specific mRNA enrichment (**Thomas et al., 2012**) to demonstrate that both genes are co-expressed in motor neurons *in vivo* (**Figure 1E**). Together, these data support spatial co-expression of *sei* and *ppk29*.

Previous studies suggested that transcriptional changes in ion channel transcript abundance could play a role in the adaptation of neurons to changes in environmental temperatures (**Marder, 2011**). Thus, as a first test of our hypothesis that these two ion channels might interact antagonistically to

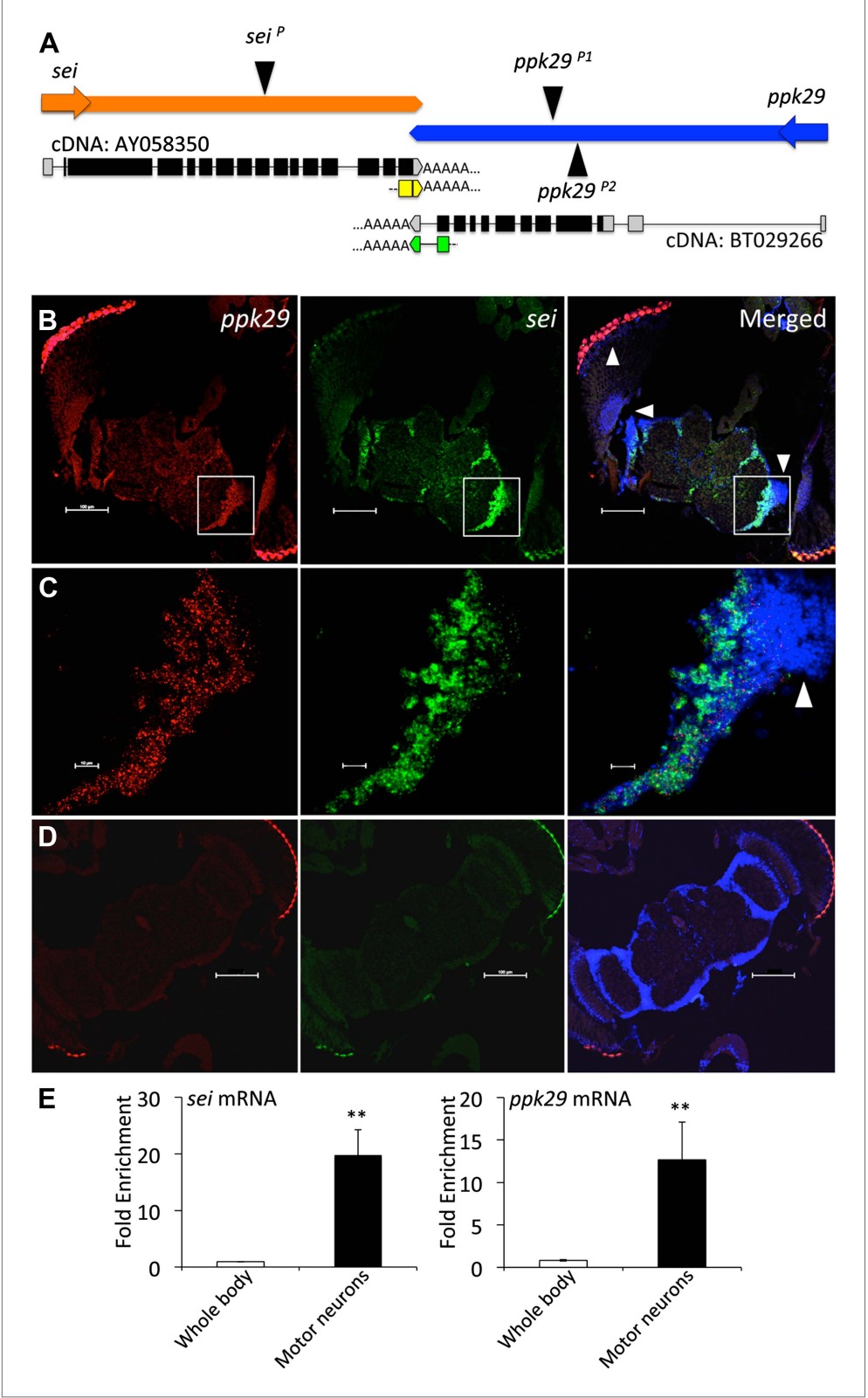

**Figure 1**. *sei* and *ppk29* are co-expressed in the nervous system. (**A**) The chromosomal architecture of *sei* and *ppk29* (2R:19,934,934- 19,944,660). Coding exons are in black. 3' and 5' untranslated regions (UTRs) are in gray. AY058350, fully sequenced *sei* cDNA; BT029266, fully sequenced *ppk29* cDNA. Black triangles represent
*Figure 1. Continued on next page*

*Figure 1. Continued*

transposons insertion sites. Arrows represent direction of transcription. Yellow boxes, *sei* 3'RACE product. Green boxes, *ppk29* 3'RACE product. (**B**) *In situ* hybridization shows *sei* and *ppk29* are co-expressed in neuronal tissues. Antisense riboprobes. Scale bar, 100 µm. (**C**) Higher magnification of white box in **B**. White arrowheads, optic lobe neurons. Red, *ppk29* signal; Green, *sei* signal; Blue, DAPI nuclear stain. Scale bar, 10 µm. (**D**) Sense riboprobe controls. Scale bar, 100 µm. (**E**) Translating Ribosome Affinity Purification (TRAP) of mRNAs from larval motor neurons shows that *sei* and *ppk29* are co-enriched in these cells relative to total body RNA. mRNA levels for each gene were measured with Real-Time qRT-PCR. N = 4 per gene. **p<0.01.

The following figure supplements are available for figure 1:

**Figure supplement 1**. *ppk29* and *sei* are co-expressed in Drosophila neuronal tissues.

regulate the neuronal response to heat we measured the relative expression levels of both genes in wild type animals that were adapted to variable environmental temperatures. In agreement with our hypothesis, we found that when animals adapted to high temperature (37°C) the transcripts levels of *sei* went up and *ppk29* went down relative to their levels at 25°C. In contrast, adaptation to colder temperature (13°C) led to an opposite effect on the expression of both genes (*Figure 2*). We conclude that both *sei* and *ppk29* are likely to play opposite roles in the regulation of neuronal activity in response to changes in ambient temperature, and that the possible interaction between these two genes is physiologically relevant.

## Mutations in *ppk29* and *sei* have opposing effects on the behavioral and neuronal responses to heat stress

The data presented in *Figure 2*, and previous reports that indicated that mutations in *sei* are highly sensitive to heat stress (*Titus et al., 1997*; *Wang et al., 1997*), led us to hypothesize that mutations in *ppk29* might lead to a protection from heat stress. Based on our model presented in Figure 6, such a protective effects for *ppk29* mutations may arise from the loss of sodium currents or alternatively due to the upregulation of SEI-dependent potassium currents. As was previously reported, we found that multiple independent mutations in *sei* lead to rapid seizures and paralysis in response to acute heat stress (*Figure 3A*). In contrast, flies carrying independent insertional alleles of *ppk29* demonstrated protection from the effects of heat stress relative to wild type and *sei* mutant animals (*Figure 3A*, *Figure 3—figure supplement 1A,B*; *ppk29*[P1] and *ppk29*[P2] are described in *Figure 1A*). These data confirmed our hypothesis that *sei* and *ppk29* play opposing roles in the neuronal response to heat stress, and are likely playing an important adaptive role in environmentally induced neuronal plasticity. We also observed contrasting behavioral responses to heat stress in animals that carry single copy insertional alleles of *sei* or *ppk29 in trans* with a chromosomal deficiency that covers both loci (*Figure 3—figure supplement 1C,D*). These data indicate that the effects of either mutation on behavior are specific and not due to other background mutations.

Previous studies indicated that the temperature-sensitive phenotype of *sei* mutants is associated with heat-induced neuronal hyperexcitability (*Kasbekar et al., 1987*). Therefore, we hypothesized that mutations in *ppk29* will lead to a hypoexcitable neuronal phenotype under heat stress. We found that the spontaneous neuronal activity of larval motor neurons is not different between *ppk29*, *sei* and wild type animals at 25°C.

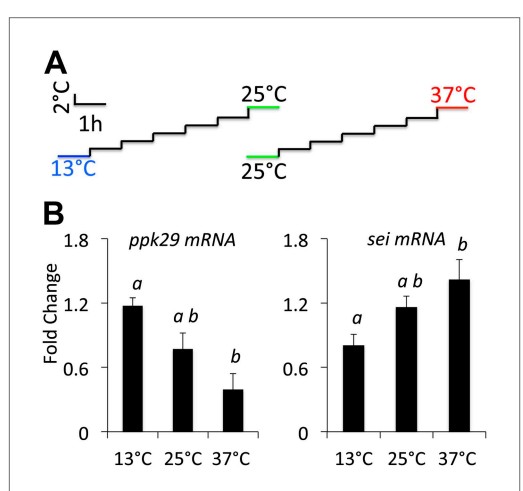

**Figure 2**. *sei* and *ppk29* transcripts are inversely regulated in response to changes in ambient temperature. (**A**) Temperature adaptation protocol. Total time from 25–37°C or 25–13°C is 7 hr. (**B**) Real-time qRT-PCR data. Different letters above bars represent statistically significant post hoc analyses (Tukey's, p<0.05, N = 4 per group).

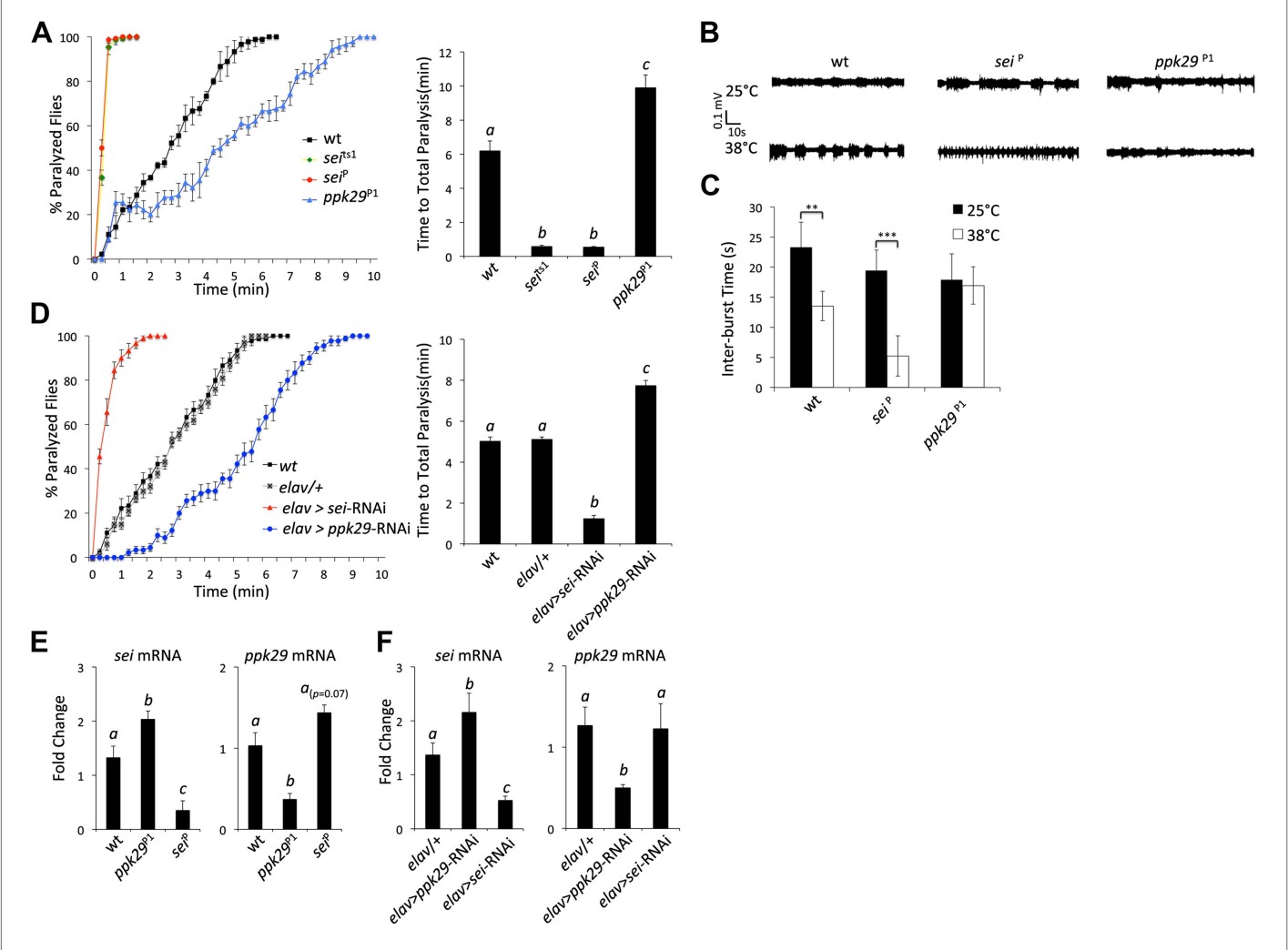

**Figure 3**. RNAi-dependent knockdowns of *ppk29* and *sei* expression lead to opposing effects on heat-induced paralysis. (**A**) The behavioral response to heat stress in *sei* and *ppk29* mutants. Left panel, cumulative paralyzed flies over time. Right panel, same data as in left panel presented as time to total paralysis (n = 16, p<0.001, one-way ANOVA). Different letters above bars represent significantly different groups (Tukey *post hoc* analysis, p<0.05). (**B**) Representative extracellular recordings from motor neurons from each genotype at 25°C and 38°C. (**C**) Summary neurophysiological data (n = 8-10 per genotype, **p<0.01, ***p<0.001, one-way ANOVA with a Tukey *post-hoc* test). (**D**) Neuronal downregulation of *sei* or *ppk29* with gene-specific RNAi constructs. Data presented as in A (n = 16, p<0.001, one-way ANOVA). (**E**) *sei* and *ppk29* mRNA levels in *sei* and *ppk29* mutant lines. Analyses were by relative real-time quantitative RT-PCR analyses. Left panel, *sei* mRNA. Right panel, *ppk29* mRNA (n = 4 per genotype, p<0.05, one-way ANOVA). (**F**) *sei* and *ppk29* mRNA levels in *sei* and *ppk29* RNAi-knockdown lines. Analyses as in E (n = 4 per genotype, p<0.05, one-way ANOVA). Data are presented as mean ± SEM. Different letters above bars represent significantly different groups (Tukey *post hoc* analysis, p<0.05).

The following figure supplements are available for figure 3:

**Figure supplement 1**. *ppk29* mutations confer protection from heat-induced paralysis.

**Figure supplement 2**. Mutations in *sei* and *ppk29* do not affect gross locomotion at room temperature.

In contrast, at 38°C wild type neurons show a small but significant increase in neuronal activity, while *sei* mutant neurons become hyperexcitable. In contrast to *sei* null and wild type animals, *ppk29* mutants are unable to increase neuronal firing rates in response to heat stress, which is consistent with a hypoexcitability phenotype (*Figure 3B,C*). We also confirmed that the larval excitability phenotypes of *sei* and *ppk29* mutants are correlated with behavior. As in our neurophysiological studies, we found that at 25°C all genotypes show normal larval locomotion (*Videos 1–3*). However, exposure to 38°C lead to

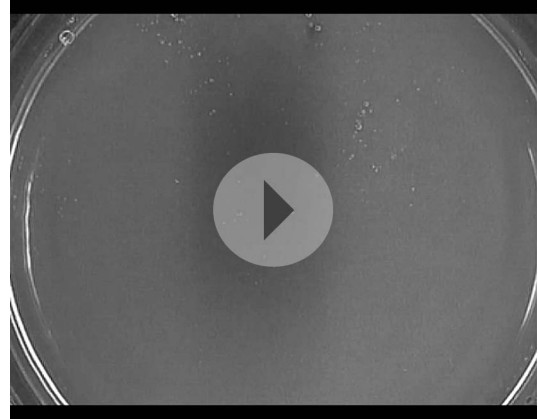

**Video 1**. Wild type larva at 25°C.

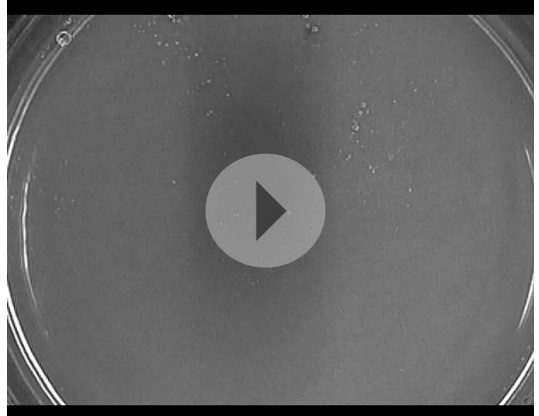

**Video 2**. *sei*[s] larva at 25°C.

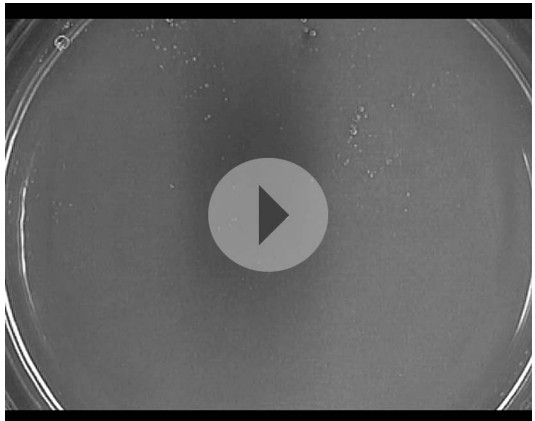

**Video 3**. *ppk29*[P1] larva at 25°C.

an abnormal seizure-like locomotion in wild type animals (twitching and rolling). This phenotype is significantly higher in *sei* mutant and RNAi knockdown larvae but completely absent in *ppk29* mutant larvae (*Figure 3—figure supplement 1E,F*; *Videos 4–6*). We conclude that *sei* and *ppk29* affect the behavioral sensitivity to heat stress via the contrasting regulation of neuronal excitability in both larval and adult stages.

Similarly to the mutant adult phenotypes, neuronal RNAi-dependent knockdown of *sei* and *ppk29* mRNAs with the neuronal *elav*-GAL4 driver lead to contrasting phenotypes that are identical to the phenotypes observed in mutants (*Figure 3D*). These data demonstrate that the observed phenotypes are neuronal-specific and suggest that quantitative changes in neuronal mRNA levels of either *sei* or *ppk29* are sufficient to induce high-sensitivity or protective phenotypes respectively. Analyses of mRNA levels in mutants and RNAi-expressing animals support the hypothesis that downregulation of *ppk29* mRNA is associated with increased *sei* mRNA levels, but the converse effect is not evident (*Figure 3E,F*). Together, these data demonstrate that the regulatory interaction between the mRNAs of *sei* and *ppk29* is not symmetric; changes in *ppk29* mRNA level downregulate *sei* mRNA, but not the other way around. We also observed contrasting phenotypes when we expressed the same gene-specific RNAi constructs in adult neurons only by using the hormonally-induced GeneSwitch version of the *elav*-GAL4 (*Osterwalder et al., 2001*; *Figure 3—figure supplement 1G*). These data show that the contrasting effects of *sei* and *ppk29* mRNA dowregulation on the neuronal response to heat stress are physiological rather than developmental. We did not observe any general locomotion defects in *sei* or *ppk29* mutants at 25°C (*Figure 3—figure supplement 2*).

Together, data presented in *Figures 2 and 3* suggest that the protective effect of mutations in *ppk29* are symptomatic in the sense that they lead to a pre-stress increase in *sei* transcript levels, which leads to a higher ability of the nervous system to deal with the acute heat stress even without prior adaptation to slow temperature increase.

## The Protective effect of *ppk29* mutations are mediated by *sei* channel activity

Although our data suggest that the contrasting heat-induced phenotypes of *sei* and *ppk29* mutants are possibly mediated via mRNA-dependent interactions, they do not exclude the possibility that the two channels also interact at the protein level. Therefore, we investigated whether the protection from heat stress in *ppk29* mutants is mediated by the loss of PPK29 channel activity or the up-regulation

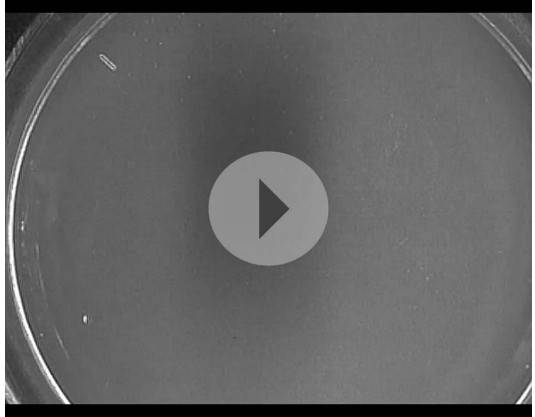

**Video 4**. Wild type larva at 38°C.

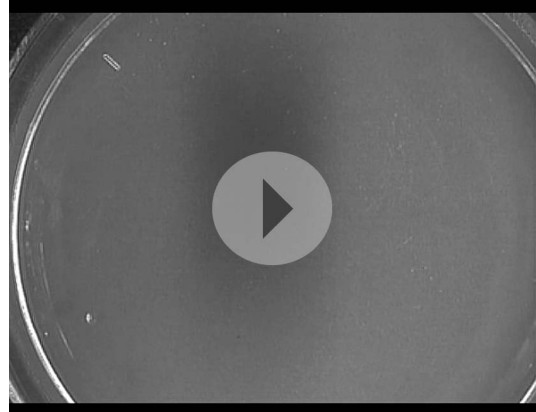

**Video 5**. *sei*[P] larva at 38°C.

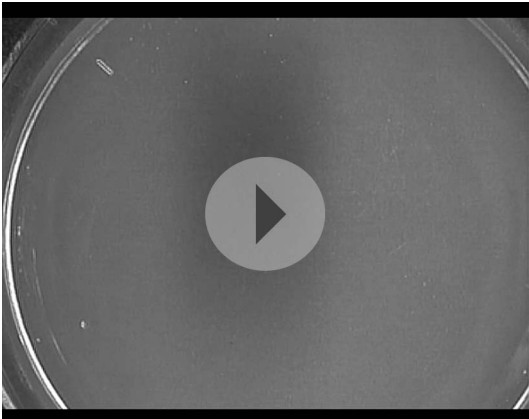

**Video 6**. *ppk29*[P1] larva at 38°C.

of *sei* mRNAs (As shown in *Figure 3E,F*). To test this we first blocked SEI channel activity in wild type and *ppk29* mutant animals by using two different hERG channel blockers (*Afrasiabi et al., 2010*). These studies reveal that blocking SEI activity in wild type animals phenocopies the heat sensitivity phenotype of the *sei* mutation, which indicate that the drugs are successfully blocking SEI channels in the fly. Similar to wild type animals, blocking SEI activity in *ppk29* mutants reduce their resistance to heat stress to a level comparable to wild type animals in a dose-dependent manner (*Figure 4A*, *Figure 4—figure supplement 1*). These data are in agreement with the expression data (*Figure 3E,F*), and strongly indicate that the *ppk29*-mediated protection from heat stress is due, at least in part, to increased SEI $K^+$ channel activity rather then the loss of *ppk29*-dependent $Na^+$ currents.

## The *ppk29* mRNA affects *sei* function by serving as a natural antisense regulatory RNA

Our hypothesis predicts that the 3'UTR of *ppk29* can regulate *sei* function by acting as a natural antisense RNA. To test directly this hypothesis we generated transgenic fly lines that can express the cDNAs of either *sei* or *ppk29* with or without their endogenous 3'UTR, or their 3'UTRs alone (*Figure 4B*) by using the UAS-GAL4 system. Remarkably, we found that the expression of the *ppk29* endogenous 3'UTR alone or the cDNA with the 3'UTR is sufficient to rescue the *ppk29* mutation. In contrast, expression of *ppk29* cDNA alone is not sufficient to completely rescue the phenotype of the *ppk29* mutation (*Figure 4C*). In agreement with the pharmacological studies, these data demonstrate that the main protective effect of *ppk29* mutations is mediated via 3'UTR-dependent regulation of SEI, independent of PPK29 channel functions. Nevertheless, we also found that a complete rescue of the *ppk29* mutation phenotype require the expression of the *ppk29* cDNA with its endogenous 3'UTR. Therefore, PPK29 channel activity may also contribute neuronal excitability independent of *sei* regulation. In addition, since the observed effects of *ppk29* transgenes on *sei* function are *in trans,* these data show that the two genes can interact at the transcript level independent of their chromosomal proximity. Unlike for *ppk29*, the neuronal expression of *sei* cDNA with or without its endogenous 3'UTR, but not the 3'UTR alone, is sufficient

to rescue the *sei* mutation (*Figure 4D*). These data further show that *sei* is the focal physiological element in the neuronal response to heat stress, and that the mRNA 3'UTR-dependent interaction between *sei* and *ppk29* is not symmetric.

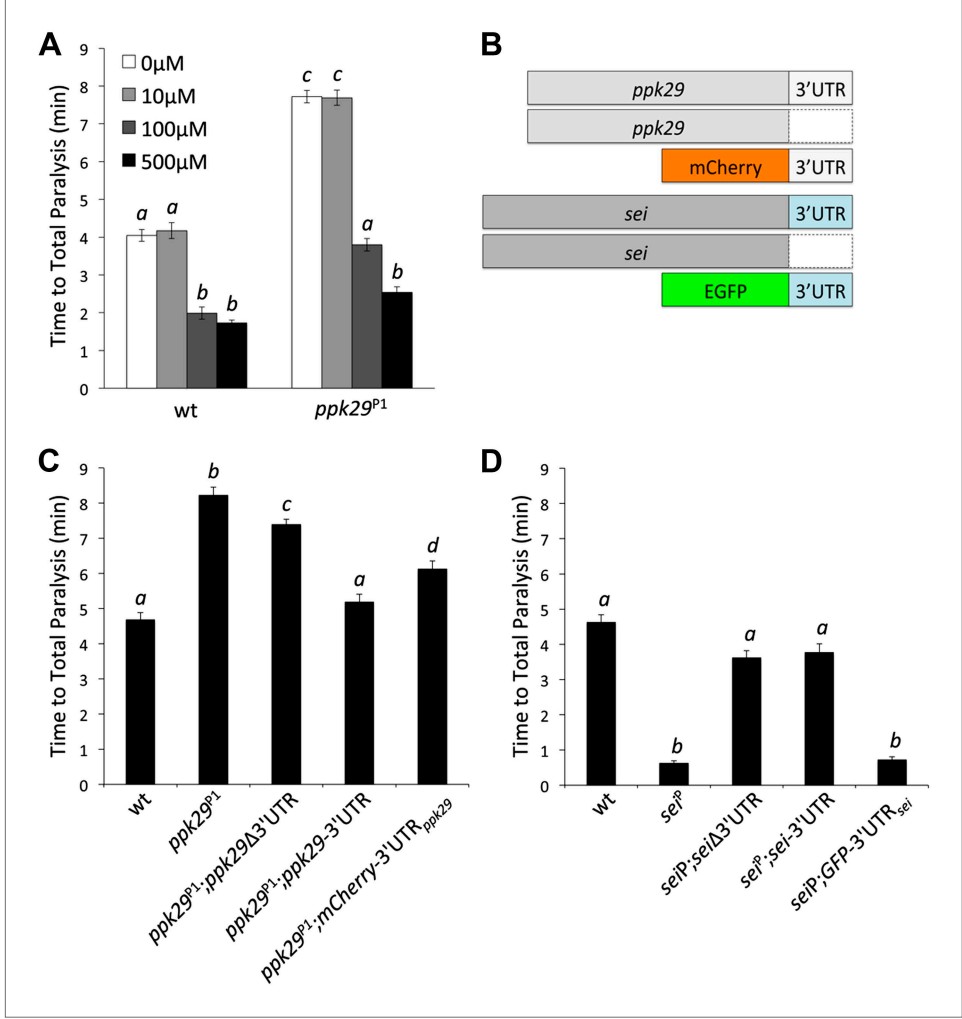

**Figure 4**. The Protective Effect of *ppk29* Mutations is Mediated by SEI Channel Activity. (**A**) Blocking SEI channel activity in *ppk29* mutants with the hERG channel blocker Cisapride eliminate the protective effect in a dose dependent manner (n = 8 per genotype, p<0.01, two-way ANOVA; genotype, dose, and genotype by dose showed significant effects, *p*=<0.001). (**B**) Schematic representation of transgenic constructs. (**C**) Neuronal expression of *ppk29*-3'UTR is sufficient to rescue the majority of the protective effect of the *ppk29* mutation (n = 12, p<0.01, one-way ANOVA). Data are presented as mean ± SEM. Different letters above bars represent significantly different groups (Tukey *post hoc* analysis, p<0.05). (**D**) Neuronal expression of *sei* cDNA with or without its endogenous 3'UTR, but not the 3'UTR alone, is sufficient to rescue the *sei* mutation (n = 12, p<0.001, one-way ANOVA).

The following figure supplements are available for figure 4:

**Figure supplement 1**. The protective effect of *ppk29* mutations depends on SEI K+ channel activity.

---

We next investigated the role of *ppk29* 3'UTR in regulating *sei* mRNA expression and heat-induced seizures and paralysis. Consistent with our model, neuronal overexpression of *sei* cDNA (with or without its endogenous 3'UTR but not the 3'UTR alone) is sufficient to protect animals from heat stress as in *ppk29* mutants (**Figure 5A**). We also found that neuronal overexpression of a *ppk29* cDNA with its endogenous 3'UTR or *ppk29* 3'UTR alone, but not the cDNA alone, is sufficient to induce heat sensitivity as in *sei* mutants (**Figure 5B**). In agreement with the behavioral data, overexpression of the *ppk29*-3'UTR is sufficient to reduce endogenous *sei* mRNA levels but overexpression of *sei*-3'UTR alone does not have a similar effect on *ppk29* (**Figure 5C**). These data demonstrate that elevated levels of *ppk29*-3'UTR alone in *trans* are sufficient to affect neuronal physiology by downregulating *sei* mRNA levels. Expression of the *ppk29* related constructs specifically in the adult nervous system by

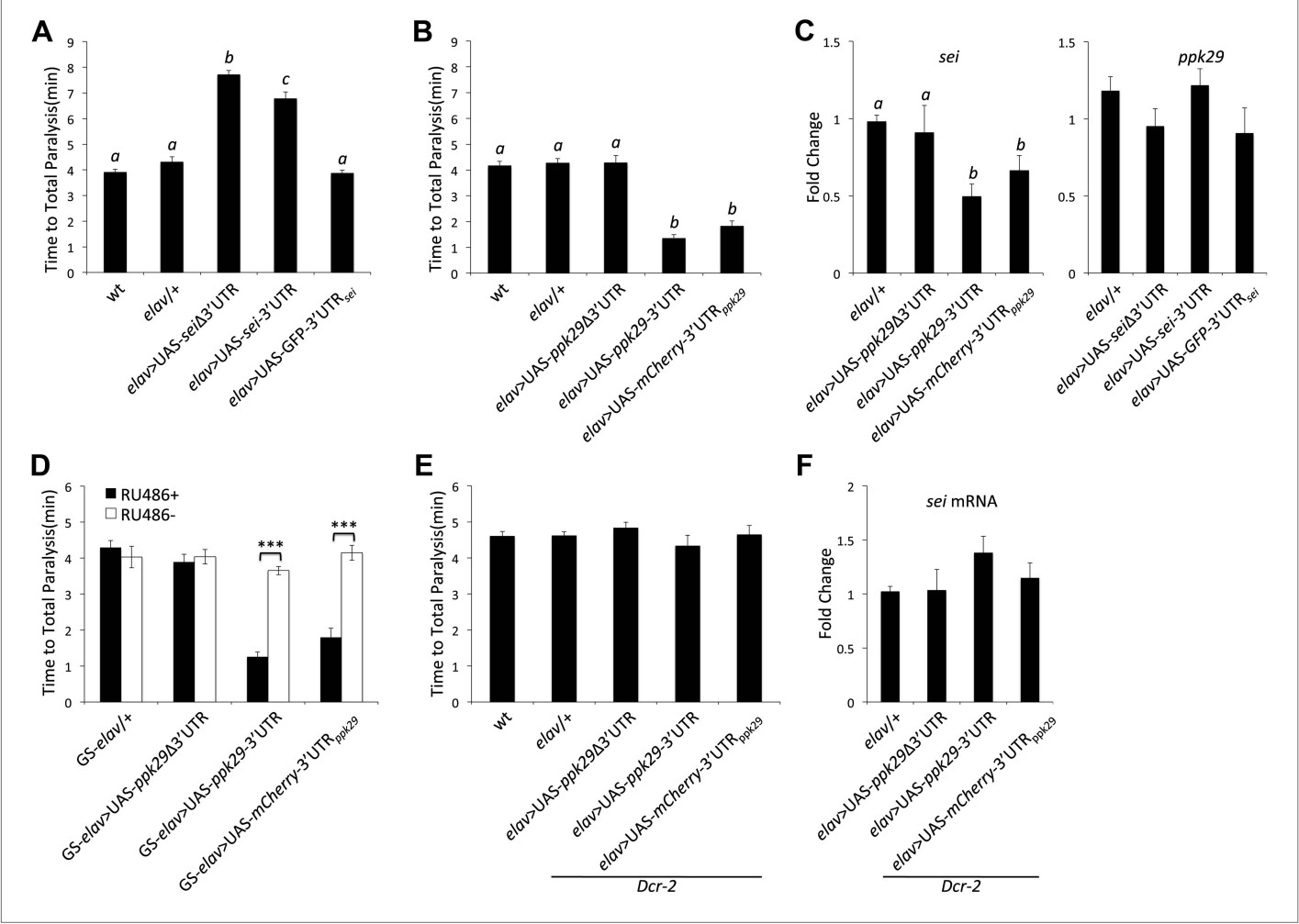

**Figure 5**. *ppk29*-dependent regulation of *sei* depends on the canonical RISC pathway. (**A**) Neuronal overexpression of *sei* cDNA with or without its endogenous 3'UTR in wild type animals leads to a protection from heat-induced paralysis (n = 12, p<0.001, one-way ANOVA). (**B**) Neuronal overexpression of the *ppk29* cDNA with its endogenous 3'UTR or the 3'UTR alone, but not the *ppk29* cDNA lone, is sufficient to induce *sei* mutant-like heat sensitivity phenotype (n = 12, p<0.001, one-way ANOVA). (**C**) Real-time qRT-PCR analyses of *sei* and *ppk29* mRNA level. Overexpression of *ppk29* cDNA with its 3'UTR or the 3'UTR alone, but not the cDNA alone, is sufficient to downregulate endogenous *sei* mRNA levels (left panel) but not conversely (right panel) (n = 4, p<0.05, one-way ANOVA). (**D**) Adult-specific neuronal overexpression of *ppk29*-3'UTR with the hormone inducible GeneSwitch *elav*-GAL4 is sufficient to induce *sei* mutant-like phenotype (n = 12, ***p<0.001; two-way ANOVA, genotype, RU486, and their interaction are significant, p=<0.001). (**E** and **F**) The effect of *ppk29* 3'UTR overexpression on heat sensitivity and sei mRNA downregulation is abolished in the *Dcr-2* mutant background (n = 12, one-way ANOVA). (**F**) Real-time qRT-PCR (n = 4, NS, one-way ANOVA). Data are presented as mean ± SEM. Different letters above bars represent significantly different groups (Tukey post hoc analysis, p<0.05).

using the GeneSwitch *elav*-GAL4 driver demonstrate that the observed effects of *ppk29*-3'UTR overexpression on *sei* function and behavior are physiological and not developmental (***Figure 5D***). Thus, our data prove that downregulation of *sei* expression leads to neuronal heat sensitivity while increased *sei* expression leads to a protection, and that the relative abundance of *sei* transcripts in neurons is affected by the expression levels of *ppk29*.

## The mRNA-dependent interaction between *ppk29* and *sei* depends on the canonical endogenous siRNA pathway

The above data show that *ppk29* mRNA can serve as a regulatory antisense RNA in addition to its capacity to encode for a DEG/ENaC subunit. The processing of endo-siRNAs depends on *Dicer2* (*Dcr-2*) in flies (***Czech et al., 2008***). Thus, we investigated whether the regulatory impact of *ppk29*-3'UTR on *sei* mRNA levels and behavior depends on the RNAi machinery. We find that in the background of

the *Dcr-2* mutation neuronal overexpression of the *ppk29*-3′UTR has no effect on the behavioral response to heat stress (*Figure 5E*) or on the expression levels of *sei* (*Figure 5F*). These data demonstrate that the regulatory function of *ppk29*-3′UTR depends on the endogenous siRNA pathway.

The finding that *ppk29* can regulate *sei* mRNA levels via the canonical siRNA pathway may also explain why the mRNA 3′UTR-dependent interaction between sei and *ppk29* is not symmetric. Recent studies of the molecular mechanism that underly the specificity of the RNAi machinery indicate that the protein complex that mediate the recognition of the target RNA by the short dsRNA are not symmetrical. Thus, via mechanisms that are not fully understood, RISC treats only one of the strands as a guide (*Tomari et al., 2004*; *Rand et al., 2005*; *Betancur and Tomari, 2012*; *Noland and Doudna, 2013*). It is likely that a similar mechanism is at play here. Our data indicate that when forming siRNA duplexes, *ppk29* 3′UTR is the preferred guide strand during RISC loading and the subsequent mRNA target identification.

## Concluding remarks

Here we describe a novel mechanism for the regulation of ion channel functions and neuronal excitability via a natural antisense mRNA (*Figure 6*). While this is a novel mechanism, it is by no means the only known RNA-dependent mechanism for the regulation of ion channel functions. For example, the double-stranded RNA helicase *maleless* (*mle*) regulates the *Drosophila* voltage-gated sodium channel *paralytic* (*para*) via A-to-I RNA editing. Mutations in *mle* lead to aberrant editing of *para*, splicing errors, and subsequent low channel activity (*Reenan et al., 2000*). Other examples include the putative transcription factor *down and out* (*dao*), which seem to affect *sei* transcription levels (*Fergestad et al., 2010*), the potassium-independent effects of the *sei*-related mammalian EAG potassium channel on cellular signaling (*Hegle et al., 2006*), and other diverse mechanisms for the co-regulation of various ion channels (*MacLean et al., 2005*; *Ransdell et al., 2013*).

It is unlikely that the type of interaction we have identified between the mRNAs of *sei* and *ppk29* is unique. Bioinformatic analyses of genome sequences show that at least two of the three fly and three out of the eight human *eag*-like (KCNH-type) channels are organized in a chromosomal architecture that is similar to that of *sei* and *ppk29* (*Table 1*). The functional diversity of the converging genes in each of the pairs we uncovered suggest that, like in the case of *sei/ppk29*, the actual protein identity is secondary to the mRNA level interactions. However, to conclusively test this hypothesis will require additional experimental molecular and biochemical analyses of these loci in the fly and mammalian systems.

Our findings also indicate that the regulatory interaction between *sei* and *ppk29* may play a role in the homeostatic response to slow changes in environmental temperature (*Figure 2*). However, our

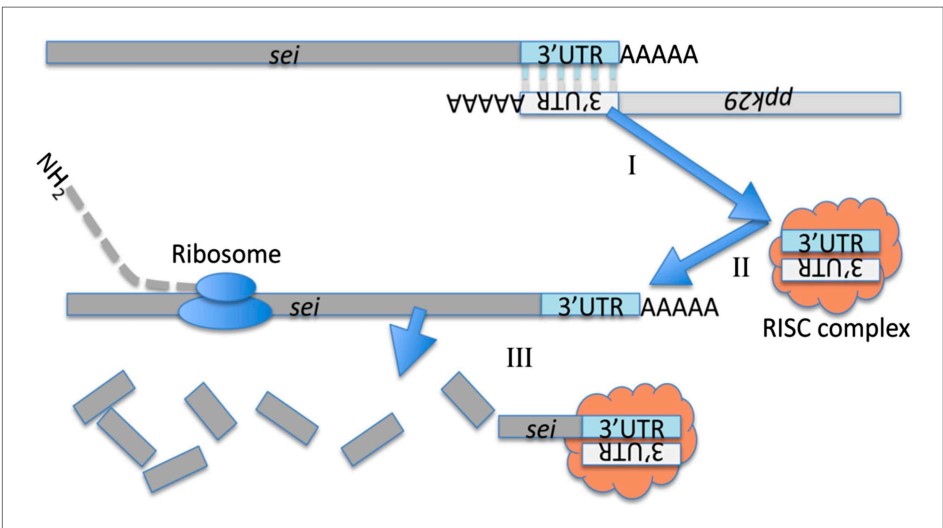

**Figure 6**. Cartoon depicting a model for the molecular interaction between *sei* and *ppk29*. The chromosomal organization of these two genes suggest they could generate endogenous siRNA by convergent transcription. (I) The complementary 3′UTRs of *sei* and *ppk29* mRNAs form a dsRNA. (II) *Dicer-2* cleaves dsRNAs into siRNAs. (III) The loaded RISC complex targets *sei* transcripts for degradation via the canonical siRNA pathway.

**Table 1.** Fly and human *eag*-like channels that are possibly regulated via convergent transcription with an unrelated mRNA

| Species | eag-like gene | Converging gene |
|---|---|---|
| *Drosophila* | *sei* | *ppk29* |
| | *eag* | *hiw* |
| Human | KCNH1 | HHAT |
| | KCNH3 | MCRS1 |
| | KCNH7 | GCA |

Please note that the converging genes are functionally diverse, which suggest that their protein identities might not play a role in their regulatory functions.

current genetic and transgenic tools make it impossible for us to completely disentangle the direct effects of temperature changes on *sei* and *ppk29* transcription and the indirect effects via the interactions of their mRNAs. Thus, more studies will be needed to further establish *ppk29* mRNA as a homeostatic factor, beyond its effects on the acute heat response.

In contrast to the linear simplicity of the 'central dogma of molecular biology' (*Crick, 1970*), we now know that the true molecular landscape of cells is complex and far from linear. In this regard, our studies provide an additional layer of regulatory complexity, and support the idea that mRNAs, which are typically thought to solely act as the template for protein translation, can also serve as regulatory RNAs, independent of their protein-coding capacity. Thus, the abundance of convergent transcription of protein-coding genes in eukaryotic genomes suggests that many other mRNAs might serve dual functions that are not necessary associated with the same cellular or physiological processes. Furthermore, although the phenomenon of mRNA-dependent interaction between the two genes we describe here occurs in *cis* (*Figure 6*), we currently have no reason to assume that similar interactions between RNAs cannot occur in *trans* as well. Consequently, it is likely that some of the evolutionary changes observed in mRNAs, including those that are considered 'neutral', should be re-evaluated in light of the possible regulatory function that some mRNAs might exert independently of the proteins they encode.

## Materials and methods

### Fly strains

Flies (*Drosophila melanogaster*) were raised on standard cornmeal-agar food at 25°C and 60% relative humidity with a 12 hr light: dark cycle. Unless stated differently, the $w^{1118}$ strain was used a 'wild type'. In our hands, the heat-induced behavior and physiology of these flies were not different from the *Canton*-S strain. The original stocks for $ppk29^{P1}$ and $sei^{P}$ were obtained from the Bloomington Stock Center (Stock No. 19016, and 21935). The $ppk29^{P2}$ stock (f04205) was from the Exelixis collection at Harvard Medical School. All insertional alleles used in our studies were backcrossed into the $w^{1118}$ background for six generation. The $sei^{ts1}$ EMS-allele was from the Ganetzki lab (U of Wisconsin). The deficiency lines *Df(2R)BSC136* and *Df(2R)BSC652* (9424 and 25742), *elav*-GAL4; *UAS-Dicer2* (25750), *UAS-ppk29*$^{RNAi}$ (27241), *elav-Gal4* (33805) ,*elav-GeneSwitch*-GAL4 (43642) and *Dicer-2* mutant (32064) were from the Bloomington Stock Center (stock no.). *UAS-sei*$^{RNAi}$ was from VDRC (v3606GD).

### Transgenic constructs

The transgene *sei*Δ3'UTR was generated by amplifying *sei* coding sequence (variant RA, NP_476713) with primers 5'-AAAAGCGGCCGCATGTCCCACAAATCTTGCGT-3' and 5'-AAAATCTAGACTAATT ATTATTATCGAACAAGTCAAGGTG-3' from cDNA clone GH12235. The transgene *sei*-3'UTR was generated by amplifying the same *sei* coding sequence plus its 3'UTR (*sei*-RA, length = 95) with primers 5'-AAAAGCGGCCGCATGT-CCCACAAATCTTGCGT-3' and 5'-AAAATCTAGATTTTCGGTTAGGACC TTTATTGC-3'. The transgene *ppk29*Δ3'UTR was generated by amplifying *ppk29* coding sequence (variant PD, NP_001097442) with 5'-AAAAGCGGCCGCATGTGGCGGAAGTCAGTA-ATG-3' and 5'-AAAATCTAGACTAACCGAAAATCATGGTCTTGA-3' from cDNA clone IP06558. The transgene *ppk29*-3'UTR was generated by amplifying the same *ppk29* coding sequence plus its 3'UTR (ppk29-RD, length = 112) with primers 5'-AAAAGCGGC-CGCATGTGGCGGAAGTCAGTAATG-3' and 5'-AAATCT AGATTGACTTGTTCGATAAT-AATAATTAGGGC-3'. The transgene GFP-3'UTR$_{sei}$ was generated by amplifying the EGFP ORF from the pEGFP-N3 vector with primers 5'-AAAAGCGGCCGCATGGTGA-GCAAGGGCGA-3' and 5'-CTTGTGCACAAATAAATAAGATTCACTTGTACAGCTCGT-CCATG-3' and the *sei*-3'UTR from cDNA clone GH12235 with primers 5'-CATGGACG-AGCTGTACAAGTGAGGC TCACTTATGCTCGCTCAATCCGAATTATCTTATTTATTTGTGCACAAGCTGTTGCGAGGCTAAAGAG-3'

and 5'-AAAATCTAGATTTTCGGTTAGGA-CCTTTATTGCTTTTCGCTCTTTAGCCTCGCAACAGCTTGTGC ACAAATAAATAAGAT-3' followed by PCR fusion of the two DNA fragments. The transgene mCherry-3'UTR$_{ppk29}$ was generated by amplifying the mCherry ORF from the pCAMBIA-1300 vector with primers 5'-AAAAGCGGCCGCATGGTGAGCAAGGGCGA-3' and 5'-TAAAGAGCGAA-AGCAATAAAGGTCTT ACTTGTACAGCTCGTCCATGC-3' and ppk29-3'UTR from cDNA clone IP06558 with primers 5'-GCA TGGACGAGCTGTACAAGTAAGACCTTTATTGCTTTTCGCTCTTTAGCCTCGCAACAGC TTGTGCACAAATAAATAAGATAATTCGGATTG-3' and 5'-AAAATCTAGATTGACTTGTTCGATAATAATA ATTAGGGCTCACT-TATGCTCGCTCAATCCGAATTATCTTATTTATTTGT-3' followed by PCR fusion of the two DNA fragments. All transgenes were verified by sequencing and subsequently subcloned into the *pUASTattB* plasmid. Each of the six individual transgenes was transformed by *PhiC31* integrase-based transgenesis into two different landing chromosomal landing sites (2L:1476459 and 3L:11837236) (*Bateman et al., 2006*).

## Adult heat-induced paralysis assay

20–30 flies (2–3 days post eclosion) were anesthetized by $CO_2$ and transferred to standard *Drosophila* vials containing fresh food for 24 hr. On test day, 10 flies (1:1 male/female ratio) were transferred to an empty polystyrene vial (Genesee Scientific, San Diego, CA) without anesthesia. Flies were allowed to recover for 10 min before vials were immersed in a 41 ± 1°C water bath (ISOTEMP105; Fisher Scientific, Pittsburgh, PA). The number of cumulative paralyzed flies was counted every 15 s until all flies were paralyzed at the bottom of the vial. The proportion of paralyzed flies and the time it takes to reach total paralysis for all 10 flies were used to generate heat-induced paralysis scores.

## Negative geotaxis assay

We used the negative geotaxis response as an assay for general locomotion as we previously described (*Lu et al., 2012*). In short, groups of ten flies were introduced into an empty vial without anesthesia. Additional empty vial was taped on top. To assay locomotion, bottom vial was tapped down lightly and the number of flies that climbed above a marked 15 cm line in 15 s was recorded.

## Larval locomotion

Feeding stage 3$^{rd}$-instar larvae were used. Each larva was briefly washed in distilled water to remove all food debris and then transferred to a 3% agar plate that was equilibrated to 25 ± 1°C or 38 ± 1°C. Recording of behavior started 3 min post introduction by videotaping animals for 2 min. The total numbers of larval side-twitching events were used to quantify larval 'seizure' like behavior (*Videos 1–6*).

## Pharmacological treatments

Stock solutions of hERG blockers were kept as 10 mM Cisapride (Sigma-Aldrich, St. Louis, MO, USA) in DMSO and 100 mM E 4301 (Alomone labs, Jerusalem, Israel) in distilled water. Working solution were made by diluting the stock solutions in in 2% (wt/vol) sucrose solution. Flies were treated in groups of 20 adults (1–2 days post eclosion, 1:1 mixed sex) in a vial containing a Kimwipe tissue paper soaked with 1 ml of the drug. Flies were allowed to feed on the drug for three days at 25°C and 60% humidity. Prior to the heat stress test, treated flies were transferred to a new vial containing standard fly food without drugs for two hours. Heat-induced paralysis was assayed as above.

## RU-486 activation of the *elav*-GeneSwitch GAL4 line

A 10 mM stock solution of RU486 (mifepristone, Sigma-Aldrich, St. Louis, MO, USA) was prepared in 80% ethanol. Then, the RU486 working solution was diluted to the final concentration (500 µM) in 2% sucrose. The drug was delivered to flies as described above. Flies were treated with RU-486 or 2% sucrose for 7 days at 25°C and 60% humidity. During the feeding period 200 µl RU486 working solution or a 2% sucrose solution control were added to each vial every 2 days. Prior to behavioral tests, flies were transferred into vials containing fresh standard fly food without drugs for two hours. Heat-induced paralysis was assayed as above.

## Extracellular electrophysiological recording

Extracellular recordings of larval segmental nerves were as previously reported (*Simon et al., 2009*). Although these neuronal bundles include both motor and sensory fibers, previous studies demonstrated that the majority of the burst firing activity patterns observed in this preparation are generated by motor neurons alone (*Fox et al., 2006*). Feeding stage 3$^{rd}$-instar larvae were dissected in HL-3 solution

containing 2.0 mM CaCl$_2$, 70 mM NaCl, 5 mM KCl, 4 mM MgCl$_2$, 10 mM NaHCO$_3$, 5 mM trehalose, 115 mM sucrose, 5 mM HEPES, pH 7.2. Segmental nerves connecting to the ventral nerve cord were left intact. We preferentially recorded from segmental nerves that innervate the anterior segments with a polished glass electrode to suck up the nerves. Neuronal signals were filtered by a high-pass filter set at 100 Hz and a low-pass filter set at 10 kHz (Clampex software package). The extracellular temperature was manipulated in the recording chamber by using a temperature-control perfusion system (Multi Channel Systems MCS, Baden-Württemberg, Germany) using the following protocol: (1) Recording of neuronal activity started once the perfusion system was stable at 25°C for at least 1 min. Neuronal spikes were recorded at baseline for 3 min. (2) To acutely raise the temperature, perfusion was turned off until it reached 38°C stabley for at least 1 min. (3) Recording at 38°C was initiated 1 min after perfusion was turned on again for 3 min.

## Real-time qRT-PCR

Total RNA from adult fly heads or whole flies was extracted with the TRIzol reagent (Applied Biosystems, Grand Island, NY). First strand cDNA pool was made from total RNA (1 µg) with random hexamere oligos *SuperScript* II reverse transcriptase (Invitrogen, Grand Island, NY) in 20 µl reacting volume. cDNA pool was diluted (1:5) in distilled water. Gene specific assays were used to quantify genes with the SybrGreen method using the PowerSYBR Green Super PCR Mix (ABI Inc., Grand Island, NY) on an ABI7500 machine (Applied Biosystems) using default parameters. Gene specific assays were designed with the PrimeTime qPCR Assay design tool (Integrated DNA Technologies). The housekeeping gene *rp49* was used as an RNA loading control as previously described (*Lu et al., 2012*). Data were transformed and analyzed according to the ΔΔCt method and are represented as relative fold differences (*Lu et al., 2012*). Primer sequences used are: *sei*-forward: 5′-TTATTCAAAGGCTGTACTCGGG-3′; *sei*-reverse: 5′-GATGCCATTCGTATAGGTCCAG-3′; *ppk29*-forward: 5′-CCTCTCAGGTATTCTTCGTTGG-3′; *ppk29*-reverse: 5′-TCGGTG-GAGATGGTATAGGTC-3′; *rp49*-forward: 5′-CACCAAGCACTTCATCCG-3′; *rp49*-reverse: 5′-TCGATCCGTAACCGATGT-3′.

## Double fluorescence *in situ* hybridization

The double fluorescence in situ hybridization in fresh brain sections was performed following published protocols (*Jones et al., 2007*). Briefly, templates for the anti-sense (AS) and sense (S) control riboprobes targeting either *ppk29* or *sei* transcripts were synthesized by PCR reactions from pUAST-*ppk29* or pUAST-*sei* plasmids with the following primers: *sei*-AS left: 5′-TAATACGACTCA-CTATAGGGCATCGATTTGATTGTGGACG-3′;*sei*-AS right: 5′-CAGTATTCGGTGC-CACATTG-3′; *sei*-S left: 5′-CATCGATTTGATTGTGGACG-3′; *sei*-S right: 5′-TAATAC-GACTCACTATAGGGCAGTATTCGGTGCCACATTG-3′; *ppk29*-AS left: 5′-TAATACG-ACTCACTATAGGGAATACGAAATGTGGCGGAAG-3′; *ppk29*-AS right: 5′-GCATTTC TTCGATGCTGTCA-3′; *ppk29*-S left: 5′-AATACGAAATGTGGCGGAAG-3′; *ppk29*-S right: 5′-TAATACG ACTCACTATAGGGGCATTTCTTCGATGCTGTCA-3′. The *sei* riboprobes were labeled by DIG (DIG RNA Labeling Kit, Roche), and the *ppk29* riboprobes were labeled by fluorescein (Fluorescein RNA Labeling Kit, Roche). Freshly dissected female brains (4–5 days old) were embedded in cryo-embedding medium (Tisse-Tek OCT, Fisher Scientific, Pittsburgh, PA). Frozen tissue were cryo-sectioned at 15 µm and fixed in 4% paraformaldehyde for 5 min. Probes were used at 2 ng/µl standard ISH hybridization buffer, 65°C overnight. Post-hybridization, tissues were blocked with TNB for 30 min followed by an incubation with a peroxidase-conjugated anti-DIG antibody in TNB buffer (1:500; Anti-Digoxigenin-POD, Fab fragments, Roche) for 2 hr to detect *sei*-specific signal. To increase signal-to-noise ratio, the Tyramide Signal Amplification system (TSA) with Horseradish Peroxidase (HRP) was used. Samples were treated for 1 hr (1:50, TSA Plus Cy3, PerkinElmer, Waltham, MA). Then, samples were transferred to 0.3% hydrogen peroxide in TNT buffer to quench HRP activity for 20 min. Subsequently, the *ppk29* antisense riboprobe was detected with a peroxidase-conjugated anti-Fluorescein antibody in TNB buffer (1:500; Anti-Fluorescein-POD, Fab fragments, Roche) for 2 hr. To amplify *ppk29* signal, samples were treated with the primary antibody in TSA signal amplification buffer (1:50, TSA Plus Fluorescein, PerkinElmer) for 1 hr. Tissue sections were mounted with Vectashield mounting medium with DAPI and imaged with a confocal microscope.

## 3′RACE

The FirstChoice RLM-RACE Kit (Life Technologies, Grand Island, NY) was used to characterize the 3′UTRs of *sei* and *ppk29* by following manufacturer's instructions. Total RNA was isolated from mixed adults and 5 µg total RNA was used for first strand cDNA synthesis. Gene specific primers for PCRs were:

*ppk29*: 5'-ACTTGCGACTGCTCTCTATTC-3'; *sei*: 5'-AAACTGCACAGGGACGATTT-3'; 3'RACEOuterPrimer: 5'-GCGAGCACAGAATTAATACGACT-3'. Positive PCR products were sequenced from both ends with the PCR primers.

## Motor neuron mRNA profiling

Translating Ribosome Affinity Purification (TRAP) was used to isolate mRNAs specifically from larval motor neurons according to a recently published protocol (*Thomas et al., 2012*). In short, a GFP tagged version of the ribosomal protein *RpL10A* was specifically expressed in larval motor neurons with the motor-neuron specific driver OK6-GAL4 (*Aberle et al., 2002*; *Xiong et al., 2010*). Total RNA was extracted using the TRizol reagent (Life Sciences, Grand Island, NY) from 35 3$^{rd}$ instar larvae. Enrichment for *sei* and *ppk29* transcripts in motor neurons was measured with Real-Time qRT-PCR in TRAPped mRNAs by comparing enriched vs total RNA from the *OK6-Gal4>UAS-GFP::RpL10A* geno-type. Real-time qRT-PCR was performed as described above.

## Statistical analyses

All quantitative behavioral, molecular, and neurophysiological data were analyzed using the most recent version of the SAS package (SAS Inc.). One-way and Two-way ANOVAs were used to analyze parametric data followed by a Tukey *post hoc* analyses ($p<0.05$) when comparisons between individual groups were required. Data distributions are presented as error bars that denote Standard Error of the Mean.

## Acknowledgements

We thank members of the Ben-Shahar and DiAntonio labs for useful comments on the manuscript, the Bloomington Stock Center for fly strains, and Paula Kiefel for assistance in generating transgenic flies.

## Additional information

### Funding

| Funder | Grant reference number | Author |
| --- | --- | --- |
| National Institutes of Health | DC010244 | Yehuda Ben-Shahar |
| National Science Foundation | 1322783 | Yehuda Ben-Shahar |
| The Esther A & Joseph Klingenstein Fund | | Yehuda Ben-Shahar |

The funder had no role in study design, data collection and interpretation, or the decision to submit the work for publication.

### Author contributions

XZ, Conception and design, Acquisition of data, Analysis and interpretation of data, Drafting or revising the article; VV, Acquisition of data, Analysis and interpretation of data; ADA, Conception and design, Drafting or revising the article; YB-S, Conception and design, Analysis and interpretation of data, Drafting or revising the article

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
