## [Decision Letter]

[Editors’ note: although it is not typical of the review process at *eLife*, in this case the editors decided to include the reviews in their entirety for the authors’ consideration as they prepared their revised submission.]

Thank you for sending your work entitled “Natural Antisense Transcripts Regulate the Neuronal Stress Response and Excitability” for consideration at *eLife*. Your article has been favorably evaluated by a Senior editor and 2 reviewers, one of whom, Leslie Griffith, is a member of our Board of Reviewing Editors.

The Reviewing editor and the other reviewer discussed their comments before we reached this decision, and the Reviewing editor has assembled the following comments to help you prepare a revised submission:

The full text of the original reviews is attached for your consideration. It would be good to address as many of the comments as you can, but the 3 key issues are enumerated below.

The consensus of the reviewers was that the paper was very novel but there were several things that should be addressed to make the argument for the proposed mechanism stronger. In addition, perhaps because of the novelty of this finding, it was felt that some of the conclusions should be tempered by acknowledgement that there might be other potential mechanisms to explain the findings. Specifically:

1) Since the identification of the antisense region was based on the (sometimes imperfectly) annotated genome, the authors should isolate the mRNAs and sequence them to verify that the UTRs from the annotation are present in the actual transcripts.

2) The data on colocalization of the *ppk* and *sei* transcripts is very weak and the figure hard to interpret. A better demonstration of coexpression would substantially strengthen the argument for example, cell specific expression profiling, high quality in situ, etc.

3) Authors should include the other interpretations and caveats of the results. Specifically, the UAS-FP-3'UTR-*ppk29* thermoprotective phenotype and *sei* transcript reduction suggests the antisense mechanism, but does not prove it conclusively. The 3'UTR might have additional targets, and the evidence for this mechanism in the endogenous rather than over-expression case is still weak.

*Reviewer*
*#1:*

This is a very novel study demonstrating temperature/stress-dependent regulation of seizure via an antisense transcript that is part of a neighboring gene. The importance of this mechanism for regulation of CNS function is made clear by the behavioral outcomes.

The data seem very clear and consistent. There is one thing lacking, however, and that is information on where and how temperature acts transcriptionally. The model would suggest that transcription of *ppk* is regulated in a temp-dependent manner, but *sei* is not (its mRNA levels are controlled exclusively posttranscriptionally). Some measure of temperature effects on transcription rate or nascent transcripts would add substantially to the story and make it completely solid that the NAT is the major regulator and not some transcriptional effect of temperature on *sei*.

*Reviewer*
*#2:*

The authors show that the sodium channel ppk29 and the potassium channel sei are transcribed on opposite strands from mRNA with overlapping 3'UTRs. Reduced expression of *sei* makes flies more vulnerable to heat-induced paralysis, while reduction of ppk29 makes them more resistant. The authors propose that the 3'UTR of *ppk29* targets the *sei* transcript for destruction via the endogenous RNA interference pathway (siRNA and RISC complex).

The straightforward hypothesis is that the amount of ion exchange through neural membranes could be achieved by co-regulating the protein expression levels of sodium and potassium channels. The surprise here is that the 3'UTR of one channel (ppk29) seems to affect the mRNA levels of the other, so the regulation might be at the transcriptional stage.

This is a very interesting concept and, if supportable, makes us think about how the balance of ion channels is achieved in new ways. Because it is such a surprising and potentially impactful finding, it needs to be extremely well supported, and I think it still falls short.

1) The overlapping transcripts of *ppk29* and *sei* are suggested by computational annotation of the *Drosophila* genome. Is there any independent verification that the identification of the 3'UTRs is correct? RT-PCR, with polyA primers and *sei* or *ppk29*-specific 5' primers, should amplify the actual mRNA transcripts and confirm that the 3' UTRs exist as predicted. Do full-length cDNAs exist for these genes already, and do they support the transcript annotation?

2) For the proposed mechanism to work, *sei* and *ppk29* do have to be precisely co-expressed, and I do not find any of the data presented convincing on this point. The cell lines are next to useless, whole CNS or whole tissue profiling is at best suggestive, and the in situ figure, which would be the most helpful is too low resolution and dim to be interpretable. Adult brain in situs in flies are notoriously hard, so I applaud the authors for trying, but Figure 1 does not persuade me that these two transcripts are co-expressed. In addition, my reading of the Methods using the Tyramide amplification system and the HRP quenching makes me nervous that there may be artifactual cross-talk as well.

3) A lot of these data rest on the “thermo-protective” phenotype of *ppk29*, and this is a new result, so I would like to see a bit more about how it is measured and how variable it is.

4) The temperature-sensitive paralysis occurs very quickly (a few minutes), while the affect of temperature rearing on ion channel transcript levels shown if Figure 2 is very slow (7 hr). At what speed do the authors think the natural antisense mechanism normally is acting? To set the basal levels of sei K^+^ channel expression appropriate for current environmental conditions? This should be clarified.

5) The authors use transgenes with a fluorescent protein fused to the *ppk29* or *sei* 3'UTR to assess whether the UTRs alone are sufficient. The levels of endogenous fluorescence would be a quantifiable measure of how much the expression levels are changes and this could provide an alternative measure to the qRT-PCR data. Not essential, but should be considered.

6) The result that the *ppk29* 3'UTR alone has a thermo-protective effect is certainly interesting. Is it certain that all of its effect is due to increased levels of sei? Does the UAS-sei without UTR rescue better than the version with the UTR? Presumably it would bypass the *ppk29* regulation and produce more sei. Given the chromosomal proximity, it would be hard to make a double mutant of *sei* and *ppk29*, but perhaps RNAi could be used to confirm that reducing sei modifies the *ppk29* mutant phenotype? The sei drug blockers (Figure 4—figure supplement 1) address this a bit, but RNAi would corroborate.

The authors should connect their findings to other regulatory mechanisms that have been shown to act on channel transcripts, specifically the RNA helicase NAP thought to regulate the para sodium transcript levels and RNA editing (Reenan and Ganetzky, Neuron 2000), the Dao protein proposed to regulate potassium and sodium channel levels (Fergestad and Ganetzky, PNAS 2010), the conductance-independent affects of the eag potassium channel on intracellular signaling (Hegle and Wilson PNAS, 2006), and the co-regulation of various ion channels as described in McLean and Harris-Warrick, J. Neurophys (2005), for example. The current proposal of RNA degradation of sei by ppk29 is an interesting idea, but adding some historical and intellectual context seems appropriate for the Discussion.

---

## [Author Response]

*The consensus of the reviewers was that the paper was very novel but there were several things that should be addressed to make the argument for the proposed mechanism stronger. In addition, perhaps because of the novelty of this finding, it was felt that some of the conclusions should be tempered by acknowledgement that there might be other potential mechanisms to explain the findings*.

We appreciate the consensus view that our work is novel. As described later, we now acknowledge that some of our data cannot exclude other possible explanations for our findings (see specific responses below).

*1) Since the identification of the antisense region was based on the (sometimes imperfectly) annotated genome, the authors should isolate the mRNAs and sequence them to verify that the UTRs from the annotation are present in the actual transcripts*.

We agree with the reviewer that the prediction of 3’UTRs is often imperfect. However, in the case of the two genes in question, both sei and ppk29 gene boundaries are supported by fully sequenced cDNAs as well as RNA-seq data from the *Drosophila* modEncode database. Nevertheless, as suggested by the reviewer, we performed 3’RACE for both genes, which confirmed the data obtained from fully sequenced cDNAs available at NCBI (revised Figure 1).

*2) The data on colocalization of the* ppk *and* sei *transcripts is very weak and the figure hard to interpret. A better demonstration of coexpression would substantially strengthen the argument for example cell specific expression profiling, high quality in situ, etc*.

We now further support the co-localization of *sei* and *ppk29* by demonstrating that both genes are enriched in motor neurons by using in vivo Translating Ribosome Affinity Purification (TRAP) (new Figure 1 and new paragraph in the Results section).

*3) Authors should include the other interpretations and caveats of the results. Specifically, the UAS-FP-3'UTR-*ppk29 *thermoprotective phenotype and* sei *transcript reduction suggests the antisense mechanism, but does not prove it conclusively. The 3'UTR might have additional targets, and the evidence for this mechanism in the endogenous rather than over-expression case is still weak*.

We toned down our interpretation of the data throughout the revised manuscript.

Reviewer #1:

*The data seem very clear and consistent. There is one thing lacking, however, and that is information on where and how temperature acts transcriptionally. The model would suggest that transcription of* ppk *is regulated in a temp-dependent manner, but* sei *is not (its mRNA levels are controlled exclusively posttranscriptionally). Some measure of temperature effects on transcription rate or nascent transcripts would add substantially to the story and make it completely solid that the NAT is the major regulator and not some transcriptional effect of temperature on* sei.

The reviewer is absolutely correct that it’s very hard to disentangle the effects of direct impact of temperature on *sei* expression from the effect of the interaction with *ppk29*. We are currently following up on this line of investigation as part of larger study on the role of NATs in regulating neuronal homeostasis. We’re generating new sets of transgenic reporters that will enable us to measure activity levels of both promoters independent of transcript levels in vivo. Although these studies are beyond the scope of this manuscript, we now address this caveat in the interpretation of our data: “Our findings also indicate that the regulatory interaction between sei and ppk29 may play a role in the homeostatic response to slow changes in environmental temperature (Figure 2). However, our current genetic and transgenic tools make it impossible for us to completely disentangle the direct effects of temperature changes on *sei* and *ppk29* transcription and the indirect effects via the interactions of their mRNAs. Thus, more studies will be needed to further establish *ppk29* mRNA as a homeostatic factor, beyond its effects on the acute heat response.”

Reviewer #2:

*1) The overlapping transcripts of* ppk29 *and* sei *are suggested by computational annotation of the Drosophila genome. Is there any independent verification that the identification of the 3'UTRs is correct? RT-PCR, with polyA primers and* sei *or* ppk29*-specific 5' primers, should amplify the actual mRNA transcripts and confirm that the 3' UTRs exist as predicted. Do full-length cDNAs exist for these genes already, and do they support the transcript annotation*?

See our response to Comment 1**.**

*2) For the proposed mechanism to work,* sei *and* ppk29 *do have to be precisely co-expressed, and I do not find any of the data presented convincing on this point. The cell lines are next to useless, whole CNS or whole tissue profiling is at best suggestive, and the in situ figure, which would be the most helpful is too low resolution and dim to be interpretable. Adult brain in situs in flies are notoriously hard, so I applaud the authors for trying, but*
Figure 1
*does not persuade me that these two transcripts are co-expressed. In addition, my reading of the Methods using the Tyramide amplification system and the HRP quenching makes me nervous that there may be artifactual cross-talk as well*.

We respectfully disagree with the reviewer that the cell line data are useless. These are clonal cell lines and thus represent a single population. These data are consistent with our hypothesis. As described in our response to Comment 2, we added new data in further support of the co-localization of *sei* and *ppk29* by demonstrating that both genes are enriched in motor neurons (new Figure 1 and new paragraph in the Results section).

*3) A lot of these data rest on the “thermo-protective” phenotype of* ppk29*, and this is a new result, so I would like to see a bit more about how it is measured and how variable it is*.

We show in Figure 3, and Figure 3—figure supplement 1 data for paralysis as a function of time. We use standard error of the mean to represent variability. To ensure clarity, we revised and expanded our description of the method used.

*4) The temperature-sensitive paralysis occurs very quickly (a few minutes), while the affect of temperature rearing on ion channel transcript levels shown if*
Figure 2
*is very slow (7 hr). At what speed do the authors think the natural antisense mechanism normally is acting? To set the basal levels of sei K*^*+*^
*channel expression appropriate for current environmental conditions? This should be clarified*.

With our current tools it is impossible for us to accurately compare the kinetics of NAT activity versus transcription. Based on general principles of eukaryotic transcription, the acute effects of heat on paralysis are not likely to involve transcriptional changes. We interpret our data to indicate that when *seizure* mRNA levels are high due to a mutation in *ppk29* it leads to an increase in baseline resistance to heat. Thus, we predict that the slow adaptation to heat as seen in Figure 2 is the natural process in which flies slowly become desensitized to the effects of slowly increasing temperatures by increasing *seizure* expression and lowering *ppk29* expression. We now make this distinction clearer in our Discussion.

*5) The authors use transgenes with a fluorescent protein fused to the* ppk29 *or* se*i 3'UTR to assess whether the UTRs alone are sufficient. The levels of endogenous fluorescence would be a quantifiable measure of how much the expression levels are changes and this could provide an alternative measure to the qRT-PCR data. Not essential, but should be considered*.

We thank the reviewer for this excellent suggestion. We did try to use these in vivo reporters quantitatively. However, because GFP and mCherry are highly stable proteins in this system, the fluorescent signal does not faithfully report transcript levels. Although beyond the scope of the current study, we are currently developing new transgenic reporters that will hopefully enable us to address this point directly.

6) *The result that the* ppk29 *3'UTR alone has a thermo-protective effect is certainly interesting. Is it certain that all of its effect is due to increased levels of sei? Does the UAS-sei without UTR rescue better than the version with the UTR? Presumably it would bypass the* ppk29 *regulation and produce more sei. Given the chromosomal proximity, it would be hard to make a double mutant of* sei *and* ppk29, *but perhaps RNAi could be used to confirm that reducing sei modifies the* ppk29 *mutant phenotype? The sei drug blockers (*Figure 4—figure supplement 1*) address this a bit, but RNAi would corroborate*.

As the reviewer suggested, we tried and failed to produce a double *sei/ppk29* mutant line. We agree with the reviewer that testing this hypothesis with RNAi would help. However, generating these lines is not trivial since it requires us to combine in a single genetic background elav-Gal4, UAS-*Dcr2*, UAS-sei^RNAi^ and the *ppk29* mutation. The realization that this would be a very difficult line to generate with matching congenic controls led us to use pharmacology instead.

*The authors should connect their findings to other regulatory mechanisms that have been shown to act on channel transcripts, specifically the RNA helicase NAP thought to regulate the para sodium transcript levels and RNA editing (Reenan and Ganetzky, Neuron 2000), the Dao protein proposed to regulate potassium and sodium channel levels (Fergestad and Ganetzky, PNAS 2010), the conductance-independent affects of the eag potassium channel on intracellular signaling (Hegle and Wilson PNAS, 2006), and the co-regulation of various ion channels as described in McLean and Harris-Warrick, J. Neurophys (2005), for example. The current proposal of RNA degradation of sei by ppk29 is an interesting idea, but adding some historical and intellectual context seems appropriate for the Discussion*.

As suggested by the reviewer, we now expand our Discussion to include these important studies in the context of our findings.